# Green-Synthesized Silver Nanoparticle–Assisted Radiofrequency Ablation for Improved Thermal Treatment Distribution

**DOI:** 10.3390/nano12030426

**Published:** 2022-01-27

**Authors:** Zhannat Ashikbayeva, Arman Aitkulov, Timur Sh. Atabaev, Wilfried Blanc, Vassilis J. Inglezakis, Daniele Tosi

**Affiliations:** 1School of Engineering and Digital Sciences, Nazarbayev University, 53 Kabanbay Batyr Ave., Nur-Sultan 010000, Kazakhstan; daniele.tosi@nu.edu.kz; 2Department of Information Engineering, University of Padova, Via Gardenigo 6/A, 35131 Padova, Italy; arman.aitkulov@alumni.nu.edu.kz; 3Department of Chemistry, Nazarbayev University, 53 Kabanbay Batyr Ave., Nur-Sultan 010000, Kazakhstan; timur.atabaev@nu.edu.kz; 4Université Côte d’Azur, INPHYNI, UMR7010, CNRS, Parc Valrose, 06108 Nice, France; wilfried.blanc@inphyni.cnrs.fr; 5Department of Chemical and Process Engineering, University of Strathclyde, 75 Montrose St., Glasgow G1 1XJ, UK; vasileios.inglezakis@strath.ac.uk; 6National Laboratory Astana, Nazarbayev University, 53 Kabanbay Batyr Ave., Nur-Sultan 010000, Kazakhstan

**Keywords:** radiofrequency ablation, silver nanoparticles, distributed temperature sensing, optical fiber, green synthesis, hyperthermia, minimally invasive cancer care

## Abstract

Thermal ablation therapy is known as an advantageous alternative to surgery allowing the treatment of multiple tumors located in hard-to-reach locations or treating patients with medical conditions that are not compatible with surgery. Appropriate heat propagation and precise control over the heat propagation is considered a weak point of thermal ablation therapy. In this work, silver nanoparticles (AgNPs) are used to improve the heat propagation properties during the thermal ablation procedure. Green-synthesized silver nanoparticles offer several attractive features, such as excellent thermal conductivity, biocompatibility, and antimicrobial activity. A distributed multiplexed fiber optic sensing system is used to monitor precisely the temperature change during nanoparticle-assisted radiofrequency ablation. An array of six MgO-based nanoparticles doped optical fibers spliced to single-mode fibers allowed us to obtain the two-dimensional thermal maps in a real time employing optical backscattering reflectometry at 2 mm resolution and 120 sensing points. The silver nanoparticles at 5, 10, and 20 mg/mL were employed to investigate their heating effects at several positions on the tissue regarding the active electrode. In addition, the pristine tissue and tissue treated with agarose solution were also tested for reference purposes. The results demonstrated that silver nanoparticles could increase the temperature during thermal therapies by propagating the heat. The highest temperature increase was obtained for 5 mg/mL silver nanoparticles introduced to the area close to the electrode with a 102% increase of the ablated area compared to the pristine tissue.

## 1. Introduction

Thermal ablation is a minimally invasive cancer treatment technique aiming to destroy the tumor by applying extremely high or low temperatures [1,2]. In comparison to conventional cancer treatment techniques, thermal ablation is used in cases when there are several small tumors, when a tumor is located in a poorly accessible region [3,4], when a tumor is resistant to chemotherapy, or when surgery is impractical due to the medical conditions of the patient. Furthermore, the thermal ablation procedure is characterized by lower morbidity, lower cost, and a shorter recovery time [5] as it can be handled as outpatient care. Depending on the frequency of the electromagnetic waves employed in the ablation process, thermal ablation is differentiated as radiofrequency ablation (RFA) [6], microwave ablation (MWA) [7], laser ablation [8], and high intensity focused ultrasound (HIFU) [9].

Hepatocellular carcinoma is known as one of the frequently diagnosed cancer diseases that is typically exposed to radiofrequency ablation [10,11]. Due to the enormous blood flow and filtering function, the liver is a popular location for metastatic cancers from several organs [2,12,13]. The most commonly used type of thermal ablation for liver cancer is a radiofrequency ablation, which utilizes high-frequency electrical currents. Radiofrequency ablation causes thermal damage to the tissue by depositing electromagnetic energy [14]. 

The power is delivered to the tissue by the active electrode [15] or metal-electrodes [16,17], while the passive electrode is in the form of a metallic plate attached to the skin. The ionic agitation occurs in the targeted region around an active electrode due to the higher electrical resistance of the tissue compared to the electrode causing frictional heat in the targeted area. This heat is accumulated and intensified around the active electrode because of the difference between the small area of the active electrode and the large area of the passive electrode [18].

The vital point during thermal ablation is the ability to monitor the temperature change precisely in real time to achieve safe and complete tumor treatment [19,20]. The existing thermocouples used for temperature sensing can detect the temperature at a single point and can affect the heat propagation due to their metallic nature. Another non-invasive alternative to thermocouples is magnetic resonance imaging (MRI). However, the MRI technique lacks in terms of precision and detection speed and commonly has artifacts on the image due to the physiological motions. Moreover, MRI scanning is a costly procedure and requires MRI-compatible surgical tools [21].

For several decades, fiber-optic-based sensors (FOS) have been widely used to measure the temperature during thermal ablation along with thermocouples. The interest in fiber-optic-based sensors arises from their advantageous properties, such as their minimal invasiveness, small size, biocompatibility according to ISO 10993 standards [22], low heat conductivity, and MRI compatibility [23]. 

The non-metallic composition of fiber-optic sensors prevents the sensors from electromagnetic interference and corrosion [21]. Recent achievements in the field of fiber-optic sensors made it possible to implement multi-point temperature-sensing opportunities in space by integrating the sensing elements into a single optical fiber [24,25,26]. Two approaches utilize multiplexed sensing with a high spatial resolution that is vital in biomedical applications. 

The first approach is Fiber Bragg Grating (FBG) arrays working on the principle of wavelength division multiplexing where the sensors inscribed inside the optical fiber allow monitoring the temperature change along with the fiber in multiple points by 3–10 mm spatial resolution reduction. The second method is distributed sensing based on the Rayleigh scattering, where several fibers serve as a sensor [27]. The optical frequency-domain reflectometry (OFDR) principle is used in the distributed sensing applied to Rayleigh scattering [28].

Another important factor in thermal therapy is an appropriate heat propagation during the ablation procedure. In particular, one can consider the use of innovative materials to improve the thermal distribution or controlled release of drugs in a confined area. To date, nanomaterials are widely used to advance the heat propagation properties in the tissue during the thermal ablation procedure. For example, biocompatible gold and iron oxide magnetic nanoparticles have been broadly investigated in many studies, for instance the thermal ablation procedure [29]. It was demonstrated that better heat propagation in tissues could be achieved due to their unique physicochemical properties [29].

Moreover, metallic nanoparticles injected into the ablation area can change the optical and electrical properties of the tissue. Conductive nanomaterials can decrease the electrical impedance of the tissue, which can delay the roll-off effect to ablate a larger area. The roll-off phenomenon occurs when the dissipation power falls to zero due to the rapid increase of the electrical impedance when tissue desiccates. This phenomenon limits the size of the ablation zone [29]. Subsequently, the increase of the ablated diameter, even by 1–2 cm, leads to a sudden increase of the volume necrosis (considering the tumor necrosis being spherical) [30].

Green-synthesized silver nanoparticles are gaining considerable interest in biomedical applications. The biological green method enables the synthesis of effective biocompatible nanoparticles. For example, green-synthesized AgNPs are typically biocompatible because their surface is usually coated by natural molecules [31,32]. Green silver nanomaterials synthesized from non-toxic and safe compounds are distinguished by their fascinating properties, such as excellent conductivity, colloidal stability, antimicrobial activity, etc. [33]. 

Silver nanoparticles prepared by a green method are usually capped with organic layers making them colloidally stable. The stability of metallic silver nanoparticles in biological media has been proven by many previous studies [34,35]. Slow Ag^+^ ion leaching is not harmful to normal cells—there are many studies available on this. On the other hand, slow Ag^+^ ion leaching can be harmful to pathogens [36,37,38]; thus, AgNPs are useful as tissue-impedance lowering agents and as potential antimicrobial agents after radiofrequency ablation.

At the present moment, silver nanoparticles have been tested for drug delivery [39,40,41], screening [42,43], and in cancer treatment due to their cytotoxic effects on various cancer cells [44,45,46,47,48]. Literature analysis revealed that few reports exist on the use of AgNPs for thermal ablation. For example, Zhang et al. investigated the effect of silver nanoparticles in the ultrasound treatment of lung cancer [49]. Instantaneous heat increase with rapid dissipation was observed by Thompson et al. during the photothermal therapy of cancer using silver nanoparticles [50].

In this study, the heat propagation and temperature-enhancement ability of green-synthesized Ag nanoparticles during the radiofrequency ablation of tissue was presented for the first time. For this purpose, prepared silver nanoparticles were introduced to the different areas on the tissue to validate the heat propagation in the presence of nanomaterials. Accurate temperature monitoring was conducted employing a distributed multiplexed temperature-sensing network. The sensing setup consists of MgO-based nanoparticle doped fibers interrogated through the Optical Backscattered Reflectometry (OBR) method and working on the principle of Rayleigh scattering with a 2 mm resolution on a 2-dimensional, surface-based measurement, which allows estimating the areas of cytotoxicity and thermal damage.

## 2. Materials and Methods

### 2.1. Experimental Setup

The experimental setup for the radiofrequency ablation (RFA) of tissue was designed to deliver the electromagnetic energy to the tissue through a single-tip applicator and to simultaneously monitor the change of the temperature during the thermal ablation procedure; this was achieved by employing the silver nanoparticles and distributed optical fiber-based temperature-sensing system as presented in Figure 1 with the photographic view in Figure 2. 

The setup consisted of: (1) an Optical Backscattering Reflectometer (OBR4600, Luna Technologies, Roanoke, VA, USA) utilized to interrogate the spectral change of the optical fibers during the ablation process; (2) a computer to collect the data; (3) an RFA/MWA Hybrid Generator (LEANFA S.r.l., Ruvo di Puglia, Italy) which was employed as a source of RF ablation; (4) six MgO-based nanoparticle doped optical fibers (NPDF) positioned in parallel to record the temperature change; and (5) porcine liver used as an object of ablation treated with silver nanoparticles.

An alternating electric field was introduced to the tissue using RFA Hybrid Generator through a connected active electrode (AE) with a 0.5 cm length conical tip. The dimensions of the active electrode were 16 cm in length and 3 cm in diameter. The AE was placed ex vivo into the commercially available porcine liver purchased from a local shop. The tissue was placed on the surface of the passive electrode (PE) presented as a metallic plate. 

The current was delivered at an operating frequency of 450 kHz and power of 60 W. The generator was set at a ‘safe mode on’ condition that prevents exceeding the value of tissue resistance above 800 Ω due to the mounted impedance-meter in the RF generator. The ablation procedure was conducted for 60 s with a cooling time of 60 s. Before use, the liver phantom was stabilized until 22–25 °C for several hours to reach body temperature conditions. The initial temperature was recorded using a contact thermocouple IKA ETS–D5.

The temperature-sensing network consisted of six MgO-based nanoparticle doped optical fibers (NPDF) fabricated using a modified chemical-vapor-deposition method (MCVD) [51,52]. The multiplexing distributed temperature-sensing system with sub-millimeter resolution was constructed by splicing the MgO-based nanoparticle-doped optical fiber that was 125 µm in diameter with a core diameter of 10 µm into single-mode fibers (SMF) at different lengths that varied by 1–2 cm from each other to avoid overlapping of the spectra during the measurements. 

The prepared optical fibers network was connected to Luna OBR using SMF pigtails and an optical coupler at 1 × 8. The resolution of the optical fibers was 2.0 mm. The thermal sensitivity of the fibers was 9.4 pm/°C, which was verified by the previous work of Beisenova et al. [53]. The multiplexing distributed sensing network was positioned inside the tissue with a 4 cm distance between fibers in the y-direction as shown in Figure 2. The active electrode was inserted between the third and fourth optical fibers to measure the temperature increase during the ablation, while the passive electrode was placed underneath the tissue. The positions of the optical fibers and AE were fixed for all the experiments.

Silver nanoparticles of size 30–50 nm were dispersed in 0.2% agarose solution at a concentration of 5 mg/mL and injected passively into the tissue using a syringe. We introduced 100 μL of silver nanoparticles at three different positions regarding the position of the active electrode to observe the heating properties as shown in Figure 3.

Moreover, pristine tissue and tissue treated with agarose solution only and with silver nanoparticles at the concentrations of 10 and 20 mg/mL were also tested for broader analysis.

### 2.2. Data Acquisition and Processing

The data was collected with an optical backscatter reflectometer (Luna OBR 4600, Luna Inc., Roanoke, VA, USA) allowing to monitor the distributed temperature change during RFA. The OBR sends a signal through a fiber link and then measures the reflection caused by Rayleigh scattering [54]. By registering the propagation time of backscattered light, return losses experienced by different points along the fiber length are determined. An example of the resulting spectrum is in Figure 4, which presents the measurement of the fiber link in its default state before the start of the experiments. The elevated regions represent the gains due to the six multiplexed fibers.

Each local temperature value causes the reflection spectrum of each fiber section to shift in wavelength. The OBR in the distributed sensing mode measures the shifts between separate measurements and converts them into temperature changes. In order to compute the shift associated with a particular spatial position, the portion of the spectrum around it is processed in the frequency domain. The length of this region is the gauge length, which is equal to 5 mm in our case. The step size between the positions, i.e., the spatial resolution, is equal to 2 mm.

The measurements were recorded by points located at 3.5 cm along the tip of each fiber’s spectrum (Figure 4). These points were arranged into a 2D matrix according to the physical layout of the fibers during the experiment (Figure 1). By observing the thermal changes registered by all of the positions during one ablation session, the overall progression of temperature can be monitored. Vertically, each pair of fibers was separated by 4 mm; overall, this sensing arrangement covers a surface of 38 mm × 20 mm (760 mm^2^) with 20 × 6 sensing points (120 sensing points in total), whereas each sensing point covers a surface of 6.33 mm^2^. For each point in time during one experiment, the vertical and horizontal dimensions were interpolated using a step size of 0.4 mm in order to plot thermal 2D maps with increased precision.

### 2.3. Synthesis and Characterization of Silver Nanoparticles

Silver nanoparticles (AgNPs) were synthesized by reducing a silver nitrate solution utilizing commercially available green tea as schematically presented in Figure 5. We boiled 0.3 g of green tea leaves in pre-heated 20 mL deionized (DI) water at 100 °C for 10 min. The obtained tea extract was filtered through Whatman filter paper No.1 and using a vacuum filter. The filtered solution was stored at 2 °C for further use. We dissolved 0.01 g of silver nitrate powder in 50 mL of DI water at 40 °C on a hot plate under continuous magnetic stirring. After 5 min, 500 µL of tea extract was added dropwise. The solution was stirred for 30 min at 700 rpm. The color change from colorless solution to light brownish indicated the complete reduction of silver nitrate solution to silver nanoparticles. The obtained silver nanoparticles were cleaned with DI water and freeze-dried using a Lyotrap Freeze Dryer.

Silver nanoparticles were characterized using Transmission Electron Microscope (TEM), Scanning Electron Microscope (SEM), UV-Vis Spectrophotometer, Fourier-transform infrared spectroscopy (FTIR), and X-ray Diffraction (XRD) measurements.

#### 2.3.1. Transmission Electron Microscopy

The size and shape of synthesized silver nanoparticles were observed using Transmission electron Microscopy (JEM-1400 Plus, JEOL Ltd., Tokyo, Japan.). The nanoparticle solution was placed on a carbon-coated copper grid and dried at room temperature overnight. The TEM micrograph confirmed the size of nanoparticles varying from 30 to 50 nm, and its spherical shape as can be seen in Figure 6.

#### 2.3.2. Scanning Electron Microscopy

The Scanning Electron Microscope (Auriga Crossbeam 540, Carl Zeiss NTS GmbH, Oberkochen, Germany) analysis was conducted to see the morphological shape of synthesized nanomaterials presenting the spherical shape of nanoparticles as shown in Figure 7.

The EDC analysis was conducted using a Scanning Electron Microscope to observe the composition of synthesizing nanoparticles. The obtained results validated that the synthesized nanoparticles were composed of silver atoms for 96.7% as presented in Figure 8. The presence of oxygen is an indication of limited silver oxidation.

#### 2.3.3. UV-VIS

Another approach to confirm that the obtained nanoparticles are silver nanoparticles is to conduct UV-VIS spectral analysis (UV-VIS Evolution spectrophotometer 2000, ThermoFisher Scientific, Waltham, MA, USA) with a wavelength range from 300 to 700 nm. The sample was placed in a quartz cuvette and spectra were recorded with a 1 nm resolution. DI water was used as a blank reference. According to the literature review, a strong peak at 420 nm stands for the spherical green-synthesized silver nanoparticles as can be seen in Figure 9 [55,56].

#### 2.3.4. Fourier-Transform Infrared Reflectometry (FTIR)

Fourier-transform infrared spectra measurements of synthesized silver nanoparticles were conducted in the range of 4000 and 500 cm^−1^ to identify the functional groups using the FTIR Nicolet is10 Thermoscientific instrument (ThermoFisher Scientific, Waltham, MA, USA) that is shown in Figure 10. The FTIR measurements for green-synthesized silver nanoparticles identified the visible peaks in Figure 10 at 3248.49, 2356.19, 1773.28, 1753.22, 1700.89, 1648.19, 1248.69, and 1043.44cm^−1^. The peak at 3248.49 cm^−1^ is assigned to O-H and/or N-H stretching, a peak at 2356.19 corresponds to C-H stretching, bands at 1773.28, 1753.22, and 1700.89 cm^−1^ stand for C=O stretching, a peak at 1648.19 cm^−1^ is assigned for N-H bending, a band at 1248.69 cm^−1^ stands for C-N stretching, and a peak at 1043.44 cm^−1^ corresponds to C-O stretching [33,49].

#### 2.3.5. X-ray Diffraction (XRD)

The XRD analysis was performed using the SmartLab Rigaku instrument (Rigaku Americas Corporation, The Woodlands, TX, USA ) and shown in Figure 11. The diffraction peaks observed at 2θ degrees of 27.81°, 32.16°, 38.12°, 44.3°, 46.21°, 54.83°, 57.39°, and 64.42° correspond to reflections of (210), (122), (111), (200), (231), (142), (241), and (220) planes based on the face-centered cubic structure of synthesized silver nanoparticles that matched the literature values [57].

## 3. Results

### 3.1. Thermal Maps

The experimental results collected by interpolation of the data from all six fibers using an Optical Backscattering interferometer allowed to obtain 2D thermal maps recorded on the *x*–*y* plane. To validate the heating effects of the silver nanoparticles, the nanomaterials were introduced to the tissue in three positions regarding the AE: around, center, and side. The heating outcomes were different for each case as can be seen in Figure 12A–D. 

The temperature distribution is presented in the form of thermal maps distinguished by colors, where the yellow color identifies the area heated up to 60 °C, and the blue color corresponds to the temperature heated up to 42 °C. Figure 12 contains four examples of thermal maps obtained in four experiments in different conditions, which are presented as a benchmark. Overall, 32 experiments in total were performed to verify the repeatability of the heating outcomes when nanoparticles were introduced in different locations and at different concentrations.

### 3.2. Temporal Evaluation

The cinematic view of the temperature increase during RFA when 5 mg/mL of silver nanoparticles are injected in the central area of tissue is presented in Figure 13. The temperature change in 2D is visualized for 10, 20, 30, 40, and 50 s from the video recorded during RFA (Appendix A).

### 3.3. Heating Pattern

In order to investigate the heating patterns of the silver nanoparticles over the thermal therapy procedures, the tissue was ablated at several conditions. First, the pristine tissue without any nanomaterial treatment was exposed to the RF, and we recorded the temperature change. The heating effects of injecting only agarose solution into the tissue were also observed. Finally, silver nanoparticles at concentrations of 5, 10, and 20 mg/mL were exposed to the tissue surface during RFA at three different positions (center, around the AE, and the side from AE). The heating patterns for each heating scenario are presented in Figure 14 demonstrating the maximum temperature of ablation reached over time.

Figure 15 demonstrates the comparison of the heat obtained during RFA for pristine tissue, tissue treated with agarose, and tissue with 5 mg/mL of silver nanoparticles introduced at the central region near the active electrode. The figure highlights the main temperature regions where the ablation occurs. It was validated by researchers that the temperature range between 42 and 60 °C is the optimal range of ablation temperatures leading to protein denaturation, while 95 °C is the temperature close to the vaporization of the tissue.

### 3.4. Ablated Area

The statistical analysis of the ablated area hat underwent thermal exposure at 42 °C is presented in Figure 16. There are several heating conditions differentiated by the colors in the figure. The heating parameters varied by three concentrations of silver nanoparticles—5, 10, and 20 mg/mL—at the central region close to the active electrode, around the electrode, and at the side position from the electrode. The analysis of pristine tissue and tissue injected with only agarose solution were also conducted for reference purposes.

In addition, the area of ablation exposed to 60 °C was analyzed and presented in Figure 17 similarly to the area at 42 °C.

The statistical data of the ablated area for over 42 and 60 °C are presented in Table 1.

## 4. Discussion

Figure 12 reports four thermal maps obtained during RFA for different positions of silver nanoparticles at 5 mg/mL concentration and pristine tissue. The two-dimensional thermal maps recorded on the *x*–*y* plane show the temperature rise starting from 20 °C. The obtained thermal maps show the significant heat difference for each scenario. As can be seen from Figure 12, the most ablated area at the temperature range between 42 and 60 °C was reached when the tissue was treated with 5 mg/mL silver nanoparticles at the central area close to the active electrode, thus, confirming the contribution of silver nanoparticles in the heat increase during RFA. 

Moreover, the maximum temperature of 100 °C was attained during RFA conducted employing 5 mg/mL silver nanoparticles positioned in the central region in the tissue. However, non-uniform heat propagation occurred inside the tissue, which can be explained by the heterogeneous nature of the tissue. Figure 13 demonstrates the cinematic view of the heating process during RFA for 50 s. 

The temperature increases recorded every 10 s over the thermal procedure demonstrate the rapid heat increase over 60 °C for 40 s and the cooling process started as the machine turns off by reaching the tissue resistance threshold. The machine turns off due to the roll-off phenomenon preventing injury of the tissue during thermal therapy [58,59]. Since the power dissipated during RFA is inversely proportional to the electrical impedance, as shown in Equation (1):(1)$$P=\frac{I^{2}}{Z}$$(where *P*—power, *I*—current, and *Z*—impedance), when the impedance is high in the targeted region, the power is not dissipated anymore and the liquid–vapor transition begins [60].

The study validated that silver nanoparticles dispersed in agarose solution performed better than the heating pattern of agarose by itself, demonstrating the performance of silver nanoparticles. However, the best heating case, observed for silver nanoparticles at a concentration of 5 mg/mL, was an optimal heating parameter compared to 10 and 20 mg/mL of silver nanoparticles. The heating pattern features for different concentrations depicted in Figure 14 and Figure 15 show a uniform heat distribution when the tissue was injected with 5 mg/mL of silver nanoparticles. 

Silver nanoparticles, due to their conductivity, are intended to decrease the electrical impedance of the tissue to increase the heat deposition at the ablation region. Figure 16 and Table 1 show that the highest area of 102% enhancement during RFA was achieved for the tissue treated with 5 mg/mL silver nanoparticles injected at proximity to the active electrode compared to pristine tissue and tissue treated with the agarose solution without any nanoparticles. 

Figure 16 and Figure 17 demonstrate that the largest area of ablation was achieved when silver nanoparticles were injected at the central position near the AE. Theoretically, there is a direct relationship between the increase in nanoparticle concentration and the electrical conductivity of the tissue. Therefore, the application of nanomaterials of higher concentration should lead to a higher ablation zone. However, this theory works for the homogenous tissue. 

In practice, the concentration of 5 mg/mL of nanoparticles was validated as the optimal concentration compared to 10 and 20 mg/mL, which can be explained by the heterogeneity of the ablated tissue [60,61]. Moreover, as can be seen from Figure 17, the highest concentration of nanomaterials lowers the ablation temperature over 62 °C making it possible to conclude that the high concentration of nanoparticles can be a limiting factor during thermal ablation.

The proposed sensing setup constructed to monitor the temperature change in real time employing the advanced temperature-sensing arrays during RFA complemented with silver nanoparticles allowed sensing the temperature change with a sub-millimeter resolution. This approach made it possible to measure the temperature with a sub-millimeter resolution of high scattering fibers in comparison to the 0.1 mm limit of detection of the OBR instrument. The temperature estimation over the ablation was processed by collecting the spectral data from all six multiplexed fibers presenting the temperature evolution.

## 5. Conclusions

In this work, the experimental setup was arranged to monitor, in real time, the temperature change during RFA inside the tissue at over 3.5 m in fiber length and a sensing network constructed in parallel at a 4 mm distance from other fibers. Green-synthesized silver nanoparticles were applied to the tissue to validate its heat propagation during the radiofrequency ablation of porcine liver tissue. The results demonstrated that silver nanoparticles could increase the temperature during RFA compared to the pristine tissue by increasing the ablated area by 102% and reaching a higher maximum temperature. 

According to the obtained outcome, silver nanoparticles themselves can propagate the heat causing the temperature rise compared to the cases when only agarose solvent or pristine tissue were used. However, the AgNPs at a concentration of 5 mg/mL performed better than at the 10 and 20 mg/mL concentrations. In addition, the proposed sensing system allowed for accurately monitoring the temperature change at a 2 mm resolution over 120 sensing points.

## Figures and Tables

**Figure 1 nanomaterials-12-00426-f001:**
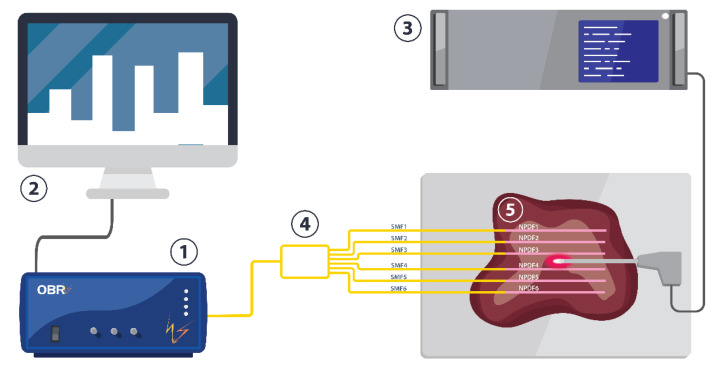
The schematic of the multiplexing setup consisting of: (**1**) the Optical Backscattering Reflectometry Luna OBR 4600; (**2**) a computer; (**3**) the RF/MW Hybrid generator; (**4**) a sensing network of 6-NPDF fibers (pink in color) spliced to single-mode fibers (SMF, yellow in color), and distributed through a fiber splitter; and (**5**) a commercially available porcine liver, used as a phantom for the RFA procedure.

**Figure 2 nanomaterials-12-00426-f002:**
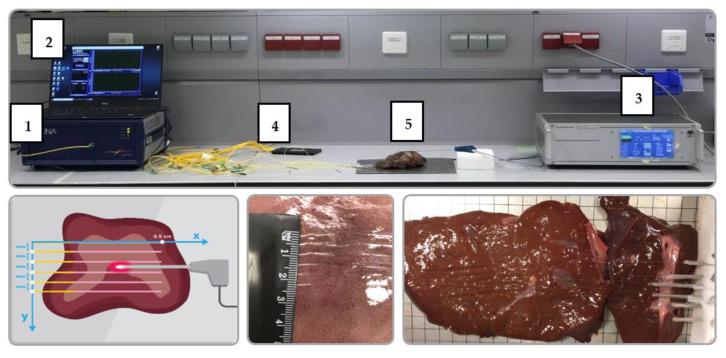
The photographic view of the experimental setup of RF ablation using silver nanoparticles and a distributed temperature-sensing system: The setup consists of: (**1**) the Optical Backscattering Reflectometry Luna OBR 4600; (**2**) a computer; (**3**) the RF/MW Hybrid generator; (**4**) a sensing network of 6-NPDF fibers; and (**5**) a commercially purchased porcine liver. The lower part of the setup presents the position of optical fibers on the surface of the porcine liver located in the *x*–*y* plane at a 4 mm distance from each other.

**Figure 3 nanomaterials-12-00426-f003:**
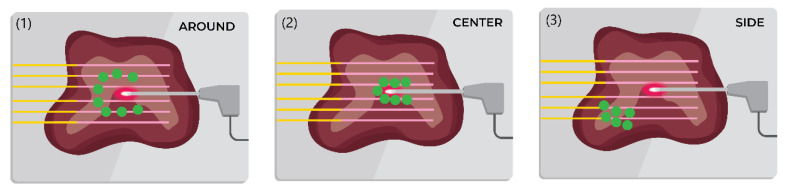
Representation of the silver nanoparticles injection position into porcine liver tissue: (**1**) around the electrode at a 1 cm distance; (**2**) center of the ablation area where the active electrode is injected; and (**3**) left side from the electrode at a distance of 1x1 cm.

**Figure 4 nanomaterials-12-00426-f004:**
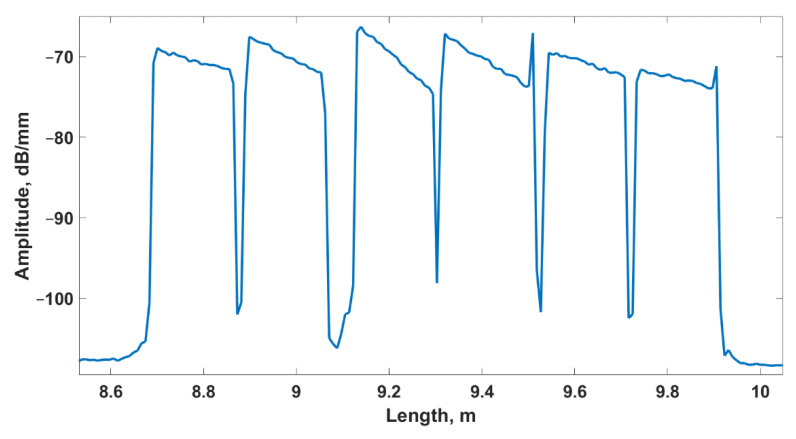
The spectrum of the multiplexed fibers in the spatial domain.

**Figure 5 nanomaterials-12-00426-f005:**
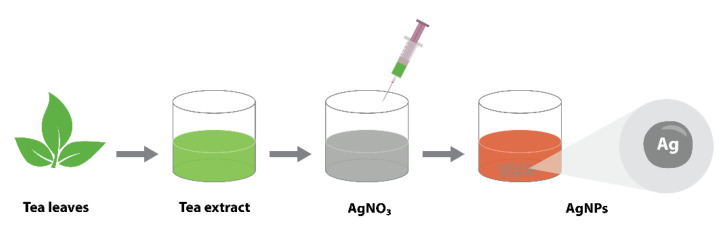
Silver nanoparticle synthesis schematics.

**Figure 6 nanomaterials-12-00426-f006:**
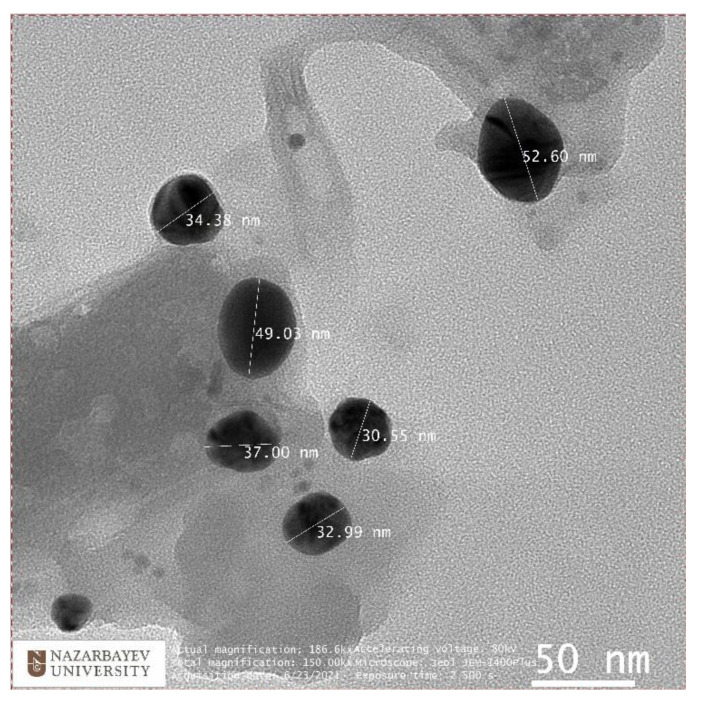
TEM image of silver nanoparticles with size indication.

**Figure 7 nanomaterials-12-00426-f007:**
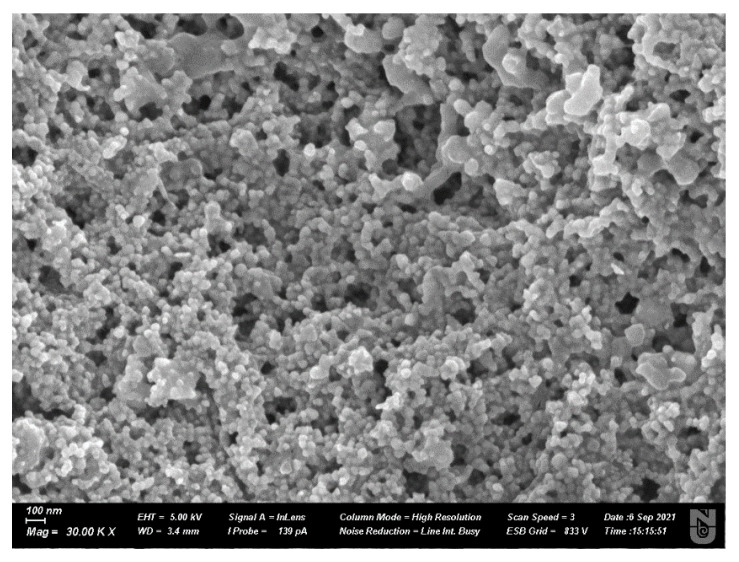
SEM micrographs of silver nanoparticles.

**Figure 8 nanomaterials-12-00426-f008:**
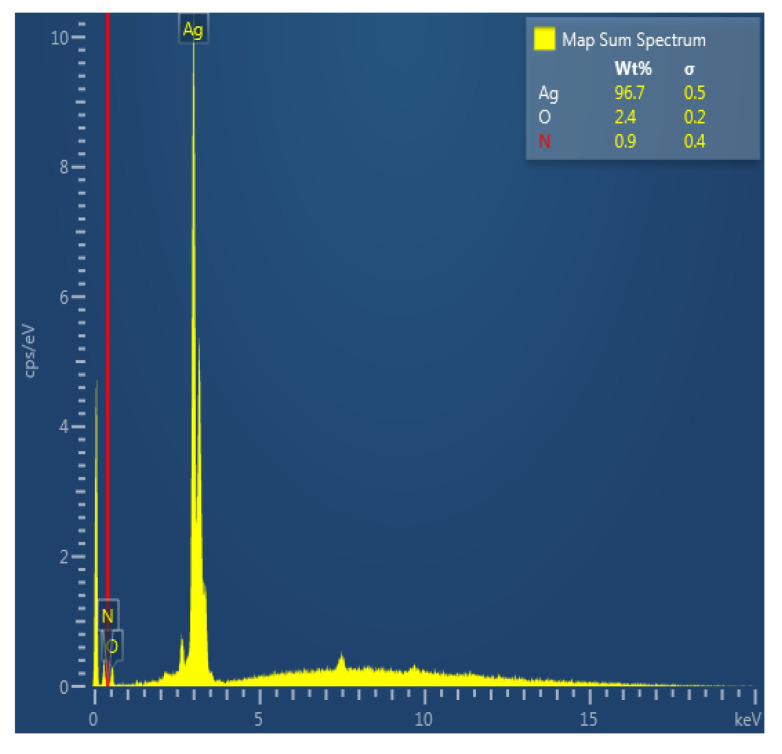
EDC composition of silver nanoparticles.

**Figure 9 nanomaterials-12-00426-f009:**
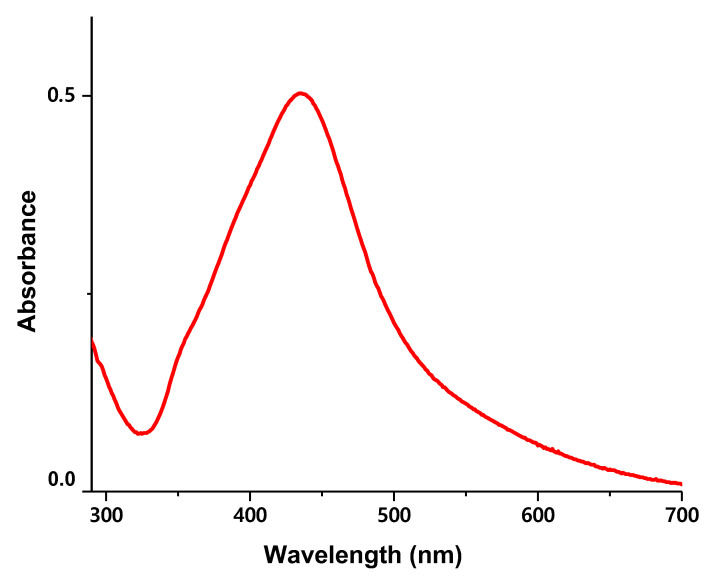
UV-VIS spectra of green-synthesized silver nanoparticles.

**Figure 10 nanomaterials-12-00426-f010:**
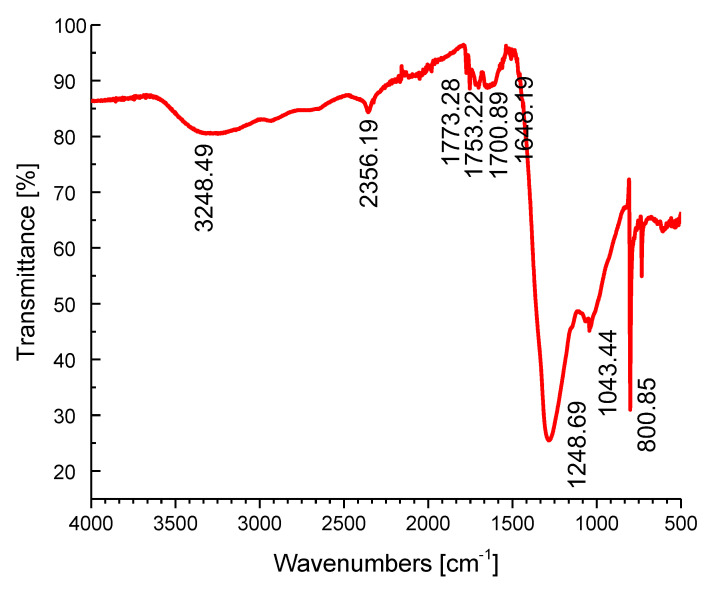
The FTIR spectral analysis of existing functional groups in green-synthesized silver nanoparticles.

**Figure 11 nanomaterials-12-00426-f011:**
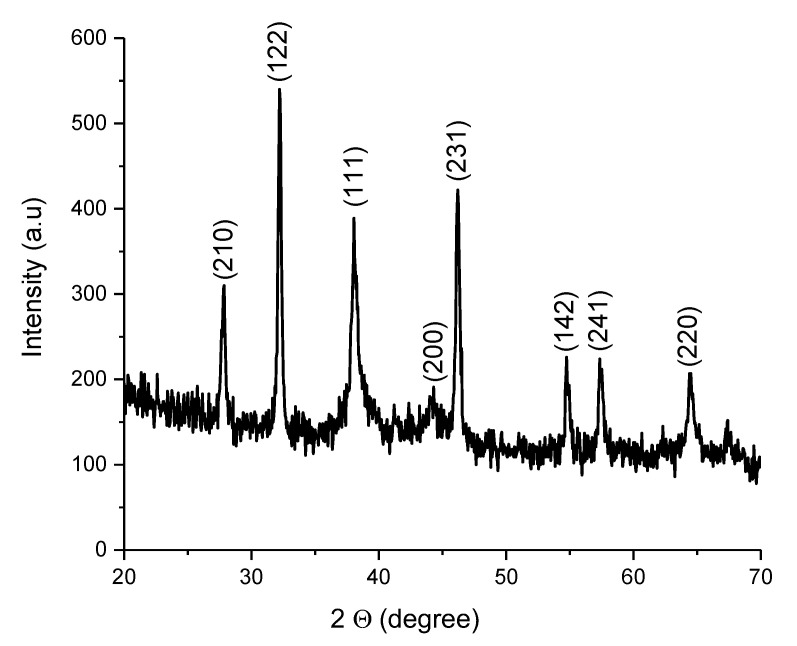
The XRD pattern of the synthesized silver nanoparticles.

**Figure 12 nanomaterials-12-00426-f012:**
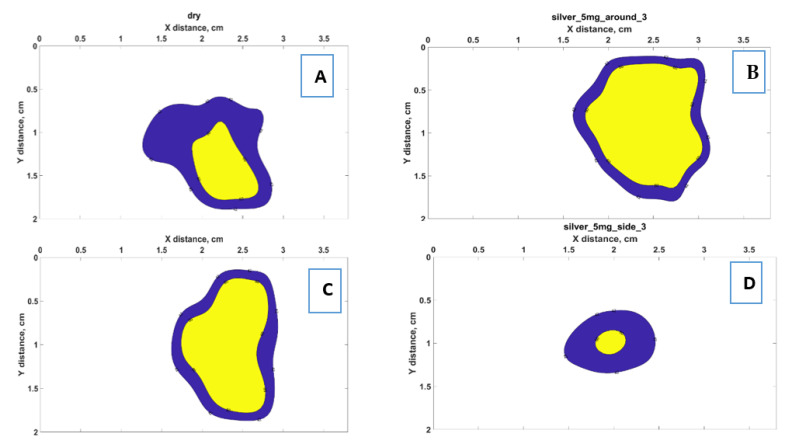
Example of thermal maps obtained for RFA in different conditions: (**A**) pristine tissue without treatment with AgNPs; (**B**) tissue with the nanoparticles injected around the electrode at a 1 cm distance; (**C**) tissue with AgNPs injected at the center at the proximity to the active electrode; (**D**) tissue with the nanoparticles injected at the side from the active electrode. The yellow zone indicates the ablated area exposed to a temperature higher than 60 °C (i.e., instantaneous thermal damage), while the blue area shows the regions exposed to >42 °C (i.e., cytotoxic regions).

**Figure 13 nanomaterials-12-00426-f013:**
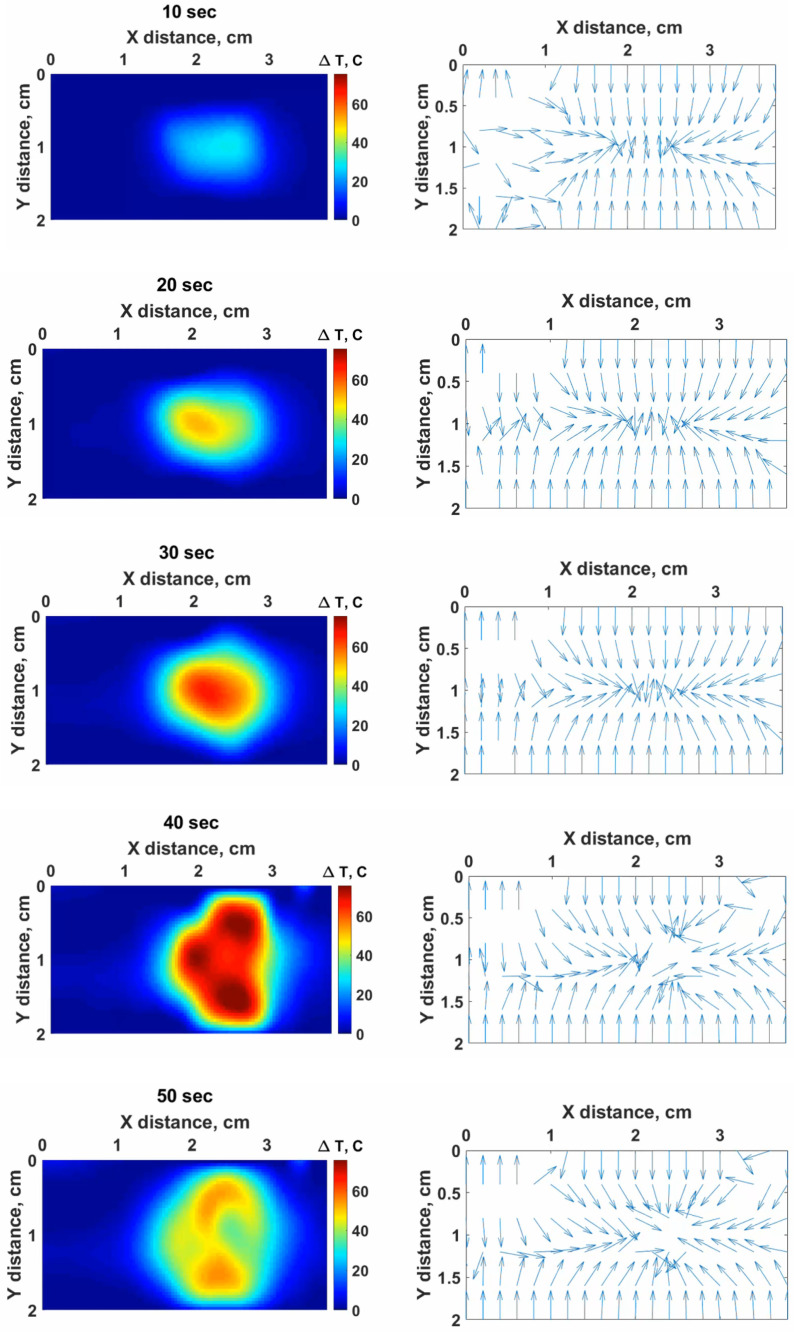
The cinematic view of the obtained 2D map during RFA at 10, 20, 30, 40, and 50 s using 5 mg/mL silver nanoparticles in the center proximity to AE.

**Figure 14 nanomaterials-12-00426-f014:**
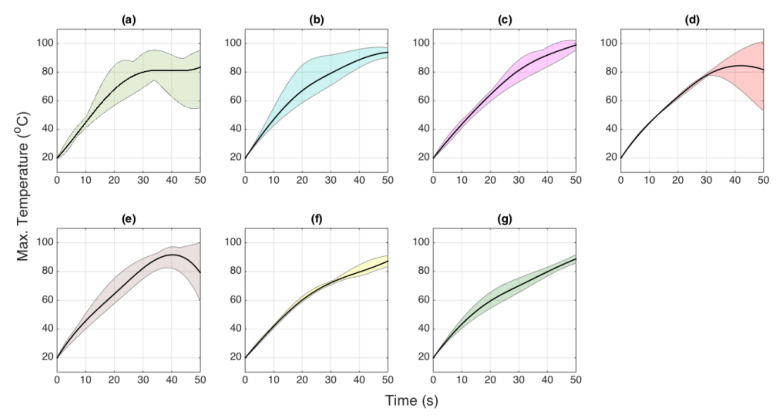
Heating patterns recorded during RFA: (**a**) Pristine tissue without nanomaterials; (**b**) agarose solution injected to the tissue; (**c**) tissue treated with AgNPs at 5 mg/mL, center position; (**d**) AgNPs at 5 mg/mL, side; (**e**) AgNPs at 5 mg/mL, around; (**f**) AgNPs at 10 mg/mL; (**g**) AgNPs at 20 mg/mL. Solid lines = average; shaded region-range between min and max of the whole experiment class.

**Figure 15 nanomaterials-12-00426-f015:**
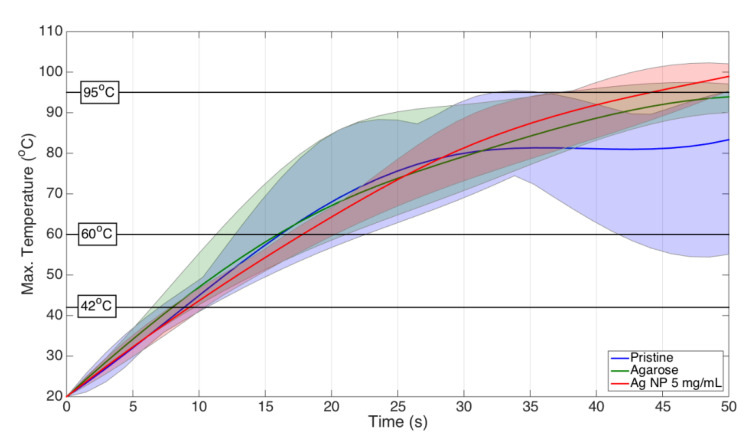
A heating comparison between pristine, agarose, and 5 mg/mL silver nanoparticles during the RFA of porcine tissue.

**Figure 16 nanomaterials-12-00426-f016:**
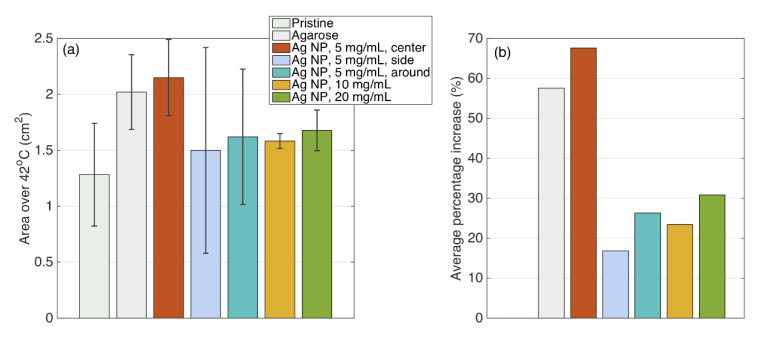
Statistical data of the area exposed to 42 °C: (**a**) For each experiment class, there is an area over 42 °C. Bars = average; errorbars = plus/minus standard deviation values. (**b**) Average increase, in %, of the 42 °C area for each experiment with respect to pristine tissue. Colors are the same on the left, while pristine tissue is 0 by definition.

**Figure 17 nanomaterials-12-00426-f017:**
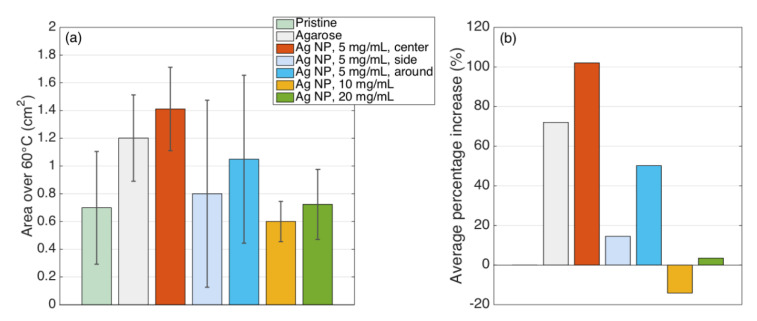
Statistical data of the area exposed to 60 °C: (**a**) For each experiment class, there is an area over 60 °C. Bars = average; errorbars = plus/minus standard deviation values. (**b**) Average increase, in %, of the 60 °C area for each experiment with respect to pristine tissue. Colors are the same on the left, while pristine tissue is 0 by definition.

**Table 1 nanomaterials-12-00426-t001:** The ablated area values for over 42 and 60 °C during RFA of the porcine liver.

Condition	Area 42 °C (cm^2^)		% Increase Over Pristine	Area 60 °C (cm^2^)		% Increase Over Pristine
	Average	Standard Deviation		Average	Standard Deviation	
Pristine	1.283	0.460	0.0	0.699	0.406	0.0
Agarose	2.021	0.333	57.6	1.201	0.311	72.0
Ag NP, 5 mg/mL, center	2.150	0.341	67.6	1.411	0.301	102.0
Ag NP, 5 mg/mL, side	1.499	0.920	16.8	0.800	0.675	14.5
Ag NP, 5 mg/mL, around	1.621	0.605	26.3	1.049	0.606	50.2
Ag NP, 10 mg/mL	1.583	0.066	23.4	0.600	0.145	−14.1
Ag NP, 20 mg/mL	1.678	0.182	30.8	0.723	0.253	3.5

## Data Availability

Data presented in this work are not publicly available at this time, but might be provided upon reasonable request to the authors.

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
