# Peer review of "Green-Synthesized Silver Nanoparticle–Assisted Radiofrequency Ablation for Improved Thermal Treatment Distribution"

_nanomaterials, 2022, doi:10.3390/nano12030426_

Round 1

Reviewer 1 Report

The thematic scope of the manuscript concerns thermal ablation therapy with the help of green-synthesized silver nanoparticles exposed to an alternating electric field. In a technical sense, the work is very well written. The results are reasonable. The manuscript deserves publication.

However, I still have a few questions:

(1) The results of the heating efficiency of the silver nanoparticles and heated areas suggest significant agglomeration of the nanoparticles in the solution. Can the authors comment further on these results?

In particular, does the external rf-electric field contribute to increasing the nanoparticle agglomeration?

Let me note that if we take magnetic nanoparticle colloidal suspension in a radio-frequency magnetic field as an example, publications suggest their increased agglomeration.

(2) Can the authors comment more on the possible distribution of silver nanoparticles after therapy?

Reviewer 2 Report

The manuscript “Green-synthesized silver nanoparticle–assisted radiofrequency ablation for improved thermal treatment distribution” explores the use of Ag nanoparticles as heat propagating agents during thermal therapies based on radiofrequency ablation (RFA). The precise control of heat and its distribution are key issues to preserve healthy tissues in in thermal ablation therapies. The authors report a 2-fold increase of thermal ablation domain in an ex vivo experiment (porcine liver) during RFA exposure compared with control tissues. The motivation of the study is well defined, however the present study requires a more comprehensive analysis that not allow to recommend it for publication in the present form.

1/ The authors inject Ag nanoparticles to enhance the thermal distribution effect due to RFA. However, bare Ag nanoparticles are known to be degraded and dissolved in the physiological medium with many examples in literature. Please provide proofs that the Ag nanoparticles maintain their metallic properties necessary for the final application after being in contact with biological media.

2/ The Ag nanoparticles are introduced at different concentrations (5, 10 and 20 mg/mL), however the optimal thermal performance is displayed by nanoparticles at 5 mg/mL. The authors should analyze and explain this effect more in detail to understand the physical mechanisms involved.

3/ In Figure 13 the heating process during RFA is recoreded for 50 seconds. A rapid increase during the first 40 s is observed and then a decrease after 50 s. The authors affirm that the machine turns off by reaching the tissue resistance threshold. Please explain in more detail this effect during the treatment.

4/ The results point out a 2-fold increase of thermal ablation domain in RFA. It is difficult to appreciate the degree of significance without a proper discussion of previous published works in the field. How the properties of the Ag nanoparticles synthesized here improve over those already reported? The choice of the Ag-base element of the nanosystem should be better inserted in the corresponding research framework.

5/ The manuscript presents grammatical errors and typo along the text. The written English has to be clearly revised.

Round 2

Reviewer 2 Report

Authors report new comments and some new data, which give answers to most of the questions raised during the first revision process. However, an answer continue to be not completely satisfactory (question 1). The authors have not provided evidence based on new data about the integrity of Ag nanoparticles after being in contact with biological medium. Experimental measurements that give electronic and structural characterization of the Ag nanoparticles in the biological medium, such as XRD, XANES or XPS should be presented. I cannot recommend the publication of the manuscript in the present form.
